# Bacterial Meningitis in Children: Neurological Complications, Associated Risk Factors, and Prevention

**DOI:** 10.3390/microorganisms9030535

**Published:** 2021-03-05

**Authors:** Abdulwahed Zainel, Hana Mitchell, Manish Sadarangani

**Affiliations:** 1Department of Pediatrics, Division of Infectious Diseases, University of British Columbia, Vancouver, BC V6t 1Z4, Canada; abdulwahed.zainel@cw.bc.ca (A.Z.); hana.mitchell@cw.bc.ca (H.M.); 2Vaccine Evaluation Center, BC Children’s Hospital, Vancouver, BC V6H 3N1, Canada

**Keywords:** bacterial meningitis, neurological sequelae, hearing loss, seizure, epilepsy, hydrocephalus, focal neurological deficit, vaccine, corticosteroid, dexamethasone

## Abstract

Bacterial meningitis is a devastating infection, with a case fatality rate of up to 30% and 50% of survivors developing neurological complications. These include short-term complications such as focal neurological deficit and subdural effusion, and long-term complications such as hearing loss, seizures, cognitive impairment and hydrocephalus. Complications develop due to bacterial toxin release and the host immune response, which lead to neuronal damage. Factors associated with increased risk of developing neurological complications include young age, delayed presentation and *Streptococcus pneumoniae* as an etiologic agent. Vaccination is the primary method of preventing bacterial meningitis and therefore its complications. There are three vaccine preventable causes: *Haemophilus influenzae* type b (Hib), *S. pneumoniae,* and *Neisseria meningitidis.* Starting antibiotics without delay is also critical to reduce the risk of neurological complications. Additionally, early adjuvant corticosteroid use in Hib meningitis reduces the risk of hearing loss and severe neurological complications.

## 1. Introduction

Acute bacterial meningitis is the most common bacterial central nervous system (CNS) infection. It is a devastating illness, especially in neonates (age < one month) and infants (age < one year). Bacterial meningitis has a high case-fatality rate of up to 30% [1,2,3,4,5], and as many as 50% of survivors develop neurological complications [1,3,6,7,8,9,10]—with outcomes highly dependent on patient’s age and the infecting organism. This article provides an overview of the short and long-term neurological complications of bacterial meningitis, the associated risk factors, and available vaccines and therapies which may reduce the risk of complications.

## 2. Epidemiology and Etiology

The incidence of bacterial meningitis in children differs by age group and is highest in infants aged younger than two months [11,12]. In the United States, the incidence rate during 2006–2007 in children under two months was 81 cases per 100,000, compared with 0.4 cases per 100,000 in children aged 11–17 years. Bacterial meningitis is more common in low and middle income countries (LMICs) compared to high income countries (HICs) [13,14]. For example, the incidence rate of meningitis in 2016 in all ages in South Sudan was 270 per 100,000 whereas in Australia it was 0.5 per 100,000 [14].

The most common organisms causing bacterial meningitis vary by age group (Table 1). Introduction of vaccines against *Haemophilus influenzae type b* (Hib)*, Neisseria meningitidis* and *Streptococcus pneumoniae* over the last three decades has led to a drastic decrease in the incidence rate of bacterial meningitis beyond the neonatal period in countries with these vaccines included as part of their routine infant and children immunization programs [15]. However, the case fatality rate has not changed significantly [16,17,18,19,20].

Hib was the leading cause of bacterial meningitis in the 1990s, but became uncommon in countries that introduced Hib immunization. However, it is still a frequent cause of bacterial meningitis in the countries were Hib vaccine is not universal or in unvaccinated children due to vaccine hesitancy [27]. In countries where Hib is now rare, *S. pneumoniae* has become the leading cause of bacterial meningitis outside the neonatal period, except in some European and Sub-Saharan African countries, where it is the second most common cause in this age group after *N. meningitidis* [26]. Group B Streptococcus (GBS) and *Escherichia coli* remain the leading causes of bacterial meningitis in neonates with little change in incidence rates over time [4,12]. Intrapartum antibiotic prophylaxis has reduced the risk of early onset, but not late-onset GBS meningitis [28,29]. Although, *Listeria monocytogenes* is an uncommon cause, it should be considered in neonates.

## 3. Pathophysiology of Bacterial Meningitis

Meningitis develops after the pathogen invades the CNS either though hematogenous route (bacteremia) or by direct extension secondary to sinusitis or mastoiditis and multiplies in the subarachnoid space. The presence of bacteria in the subarachnoid space leads to activation of the immune response, resulting in bacterial lysis. The presence of bacterial particles triggers a further inflammatory response with on-going migration of neutrophils across the blood–brain barrier and continuous cytokine and chemokine release (including IL-1B or CXCL1,2,5) (Figure 1) [7,30,31,32,33,34,35]. A persistent inflammatory state subsequently leads to decreased cerebral perfusion, cerebral edema, raised intracranial pressure, metabolic disturbances, and vasculitis, all contributing to neuronal injury and ischemia [32].

## 4. Bacterial Meningitis Complications

There are many complications that are associated with bacterial meningitis. These include short-term complications such as seizures, focal neurological deficits and subdural effusions, and long-term complications such as hearing loss, cognitive impairment, hydrocephalus, learning disability, and epilepsy [8,36,37,38]. Neurological sequelae are more likely to happen in LMICs compared with HICs [37] due to delayed presentation to medical services, lack of access to healthcare and limited resources. Additionally, complications are likely under reported in LMICs (Table 2) [7]. 

### 4.1. Short-Term Complications

#### 4.1.1. Subdural Effusion

Subdural effusions occur in 20–39% of children with bacterial meningitis [50,51,52]. Subdural effusion is most common in infants (age < one year) compared to older children [8,51]. There is no significant difference in subdural effusions complication due to Hib meningitis compared to *S. pneumoniae* and *N. meningitidis* [51]. Most subdural effusions in the context of bacterial meningitis are asymptomatic, resolve spontaneously and rarely require intervention. Indications for drainage include an infected effusion or empyema, focal neurological signs or symptoms, or increased intracranial pressure [51].

#### 4.1.2. Focal Neurological Deficit

Focal neurological deficit refers to a set of signs and symptoms resulting from a lesion localized to a specific anatomical site in central nervous system [53]. Examples include isolated limb weakness or hemiparesis, visual deficit or a speech impediment. They are estimated to occur in 3–14% of bacterial meningitis cases [41,42]. 

Acute focal neurological deficits after bacterial meningitis are usually due to ischemic stroke, but can also occur due to subdural empyema, cerebral abscess or intracranial bleeding [7]. Focal neurological deficits generally improve over months to years after the initial insult. Abscess collections can be surgically drained which usually leads to complete symptom resolution while focal deficits resulting from an ischemic event take longer to resolve. In a recent study, children with stroke following bacterial meningitis compared to children without stroke were less likely to have a normal neurological exam at discharge (21% vs. 76%) and within a seven years follow-up period (31% vs. 74%) [54]. Focal neurological deficits that persist after acute infection are not a very common complication after bacterial meningitis in childhood (Table 2); however, the incidence is highest for infections occurring in the first year of life [55].

### 4.2. Long-Term Complications

#### 4.2.1. Hearing Loss

Sensorial hearing loss is the most widely reported neurological sequelae of bacterial meningitis [7,27,44,51,56,57,58,59]. Hearing loss may develop from both the direct spread of bacterial products and as a result of the host inflammatory response in the meninges and CSF. When bacteria reach the cochlea, a severe labyrinthitis results, which leads to blood-labyrinth barrier breakage, and ultimately meningitis-associated hearing loss [7]. Around 10% of children with bacterial meningitis develop unilateral or bilateral sensorineural hearing loss [37,60]; 5% of children develop bilateral severe or profound hearing loss [37]. Hearing loss is a more common complication in infections caused by *S. pneumoniae* (14–32%), compared with *N. meningitidis* (4–23%) and *H. influenzae* (20%) [44,60]. Children with hearing loss are at risk of further developing balance disturbances [61] and speech and language delay [62], and are therefore at higher risk of having long-term behavioral problems [63].

Reversible deafness (i.e., transient hearing loss) has been documented in long term follow up of children with pneumococcal meningitis [7,61]. For example, a study done among children with pneumococcal meningitis in Bangladesh reported that 33% of children in short-term follow-up (30–40 days) had hearing loss, but only 18% had persistent hearing loss in long-term follow-up (6–24 months) following hospital. The difference was attributed to recovery of transient hearing impairment [43].

#### 4.2.2. Cognitive Impairment

Due to the irreversible neuronal damage that occurs during bacterial meningitis, the risk of developing long-term cognitive deficits and learning difficulties are significant [3,48]. The rates of cognitive impairment world-wide are difficult to estimate because there is no standardized method of measuring it and long-term data on meningitis survivors are rarely available. 

In a Dutch study, 680 children between the age of 4 and 13 years who survived bacterial meningitis were followed up for 6 years after their meningitis episode. The study followed their educational, behavioral and general health issues. The survivors were compared to a control group of healthy school-age siblings and peers, with similar socioeconomic background. It was found that 30% of children with meningitis had problems with school achievement or concentration. Additionally, these children repeated a school year twice as often as the control group (16% vs. 8%). Moreover, the post-meningitis group were referred to special-needs school four times more frequently compared to control group [48].

In a Danish nationwide population-based cohort study, adults who had a bacterial meningitis in childhood were compared to control group that included a general population of the same age and sex, their siblings and the siblings of meningitis patients. Adults who had childhood bacterial meningitis had lower educational achievements and economic self-sufficiency compared to control group. By the age of 35 years, 11%, 10.2% and 5.5% fewer had completed high school in meningococcal pneumococcal, and Hib meningitis, respectively. Additionally, 7.9%, 8.9% and 6.5% fewer had obtained a higher education in meningococcal pneumococcal, and Hib meningitis, respectively. Additionally, 3.8%, 10.6% and 4.3% had lower economic self-sufficiency in meningococcal pneumococcal, and Hib meningitis, respectively [49]. In a study done in Bangladesh, short (30–40 days) and long-term (6–24 months) follow-up revealed that 41% in both groups had deficits in mental development and 49% and 35%, respectively, had psychomotor delay [43]. Another example, in a study that was done in Brazil, 5.88% children developed learning disabilities, and 7.35% children had developmental delay [40].

Finally, psychiatric disease including anxiety and depression is likely under-recognized and underreported in meningitis survivors and contributes to cognitive difficulties and overall quality of life [64]. 

#### 4.2.3. Seizures and Epilepsy

One of the clinical presentations of bacterial meningitis is seizures [65,66]. In cases of bacterial meningitis with seizures, if seizures develop early during the illness and are easily controlled, permanent neurological complications are rarely of concern. However, if seizures are prolonged, difficult to control or develop 72 h after admission, neurological sequelae are more likely to occur and are usually suggestive of a cerebrovascular event [44,67]. In HICs, 1–5% of epilepsy cases are presumed to be due to CNS infection; including bacterial meningitis [68]. In Sub-Saharan Africa 26% of patients have epilepsy attributed to CNS infection [69].

In a neonatal bacterial meningitis study seizures have been more commonly associated with GBS then *E. coli* (41% vs. 25%) [21]. 71% of children who late seizure after bacterial meningitis had permanent focal neurological deficit [70].

#### 4.2.4. Hydrocephalus

Hydrocephalus incidence is around 7% of bacterial meningitis in children [71] and it is more common in neonates and infants; 25% [72,73]. It is more common in neonatal Gram negative meningitis [57]. Hydrocephalus may develop at the beginning of the illness or weeks later after diagnosis with bacterial meningitis. The most common type of hydrocephalus after bacterial meningitis is communicating hydrocephalus; seen in up to 52% of cases with hydrocephalus [74]. In communicating hydrocephalus CSF flows freely between the ventricles but is not adequately reabsorbed back into the blood stream. Depending on the size of hydrocephalus and resulting neurologic impairment temporary or permanent ventricular shunt placement may be required [75].

## 5. Risk Factors

There are many risk factors associated with neurological complications in bacterial meningitis (Table 3).

In general, infants are at higher risk of developing neurological complications compared to older children [3,9,23]. 71% of infants (aged < one year) with bacterial meningitis develop neurological complications compared to 38% in children aged one to five years and 10% in those aged six to 16 years [9]. Children younger than 12 months at time of diagnosis with bacterial meningitis are at increased risk of developing hydrocephalus, subdural effusion, seizure disorder and hearing loss [76]. Altered level of consciousness is associated with poor prognosis [6,9]. 82% of children with bacterial meningitis who developed neurological complications had altered level of consciousness on presentation; where, 39% of children with bacterial meningitis who did not develop neurological complications had altered level of consciousness [9]. The longer the duration that the child was unconscious, the worst the outcome is. 

In bacterial meningitis, delayed presentation to hospital increase the risk of subdural effusion, hydrocephalus, hearing impairment and seizure disorder [76]. Although delayed presentation is one of the known risk factors for developing neurological complications, there is no universal definition for the duration of the delay. In one study, children admitted with duration of illness <48 h had a lower incidence of neurological complication (40%) compared to children who were admitted after 48 h of illness [9]. Children with *S. pneumoniae* meningitis have a higher risk of developing neurological complication (75% of *S. pneumoniae* meningitis cases) compared to *N. meningitidis* (25%) and Hib (20%) [9]. *S. pneumoniae* compared to *N. meningitidis* and Hib is associated with higher risk of symptomatic seizures, hydrocephalus, hearing loss and mental retardation [76]. Delay in starting antibiotics beyond 24–72 h has a poor prognosis and leads to increased risk of severe neurological complication such as hydrocephalus, subdural effusion, hearing loss, and seizure disorder [76]. In summary, young age, delayed presentation and *S. pneumoniae* as an etiologic agent were associated with increased risk of neurological complications in both HIC and LMIC settings [11,15,54,71,76,77,78].

## 6. Prevention of Neurological Complication

### 6.1. Primary Prevention

The most effective prevention of neurological complications from bacterial meningitis is preventing the infection though infant and childhood vaccination programs. Despite the development of multiple vaccines against the organisms causing bacterial meningitis, there continue to be many meningitis outbreaks caused by vaccine-preventable organisms [79,80]. There are currently vaccines against 3 of the organisms that cause bacterial meningitis: Hib, *N. meningitidis* (capsular groups A, B, C, W and Y) and 23 of the >90 serotypes of *S. pneumoniae* [4,15]. Hib conjugate vaccine targets only type b *H. influenzae*, and is given as three or four doses before 18 months of age [81]. There are two types of vaccine against *N. meningitidis*: Conjugate vaccines against capsular groups A, C, W, and Y and protein vaccines against group B. There are two types of vaccines against *S. pneumoniae*: Pneumococcal conjugate vaccines (PCV 10 against 10 serotypes, PCV 13 against 13 serotypes) and polysaccharide vaccine against 23 serotypes which is not routinely used in healthy children [82].

Routine vaccination can lead to development of community protection by indirect effect prevention of transmission within a population [83]. Since the introduction of pneumococcal conjugate vaccines (PCVs), the overall incidence of invasive pneumococcal disease (IPD) has dropped significantly, including in unimmunized children, highlighting these indirect effects [84,85]. For example, in South Africa, Morocco, Gambia, Mozambique, Kenya and Burkina Faso, 32–81% reduction in IPD has been reported after PCV introduction, with highest reduction in children aged under 24 months (55–89%) [86]. However, infections caused by non-vaccine serotypes infections have increased in some countries. In the United States, the proportion of IPD caused by non-vaccine serotype increased from 6% to 38% after the introduction of PCV7 vaccine; sometimes referred to as serotype replacement [85,87,88]. Overall, IPD incidence dropped.

The highest rate of meningococcal disease worldwide is in the Sub-Saharan Africa, specifically in the “meningitis belt” region where major epidemics occur every 5–12 years [89]. After the introduction of MenA vaccine to the Sub-Saharan Africa in 2010, there has been a 99% reduction in group A meningitis in this region [90] and capsular group W is the currently the commonest [91]. Rate of meningococcal meningitis are much lower in other parts of Sub-Saharan Africa, although longitudinal surveillance outside of the meningitis belt is limited [92]. Following wide-spread introduction of Hib vaccine, *S. pneumoniae* accounted for 65% of acute bacterial meningitis cases in Malawi, while the rates of *N. meningitidis* have remained constant at <5% [90].

### 6.2. Secondary Prevention of Complications

#### 6.2.1. Antibiotic Therapy

It is important to have a high clinical suspicion of bacterial meningitis and start appropriate treatment without delay [93,94]. The empiric antibiotic choice should be based on the most likely causative agent for patient’s age [73,95,96]. In children, third-generation cephalosporins, such as cefotaxime or ceftriaxone, are the usual empirical choice to cover the most common organisms—*S. pneumoniae* and *N. meningitidis*. Ampicillin should be added to cover *L. monocytogenes* in very young children; some guidelines recommended for younger than 3 months, others recommended this for those younger than 1 month [95,96]. 

#### 6.2.2. Corticosteroids

Neuronal damage due to acute bacterial meningitis is not only due to bacterial invasion to the subarachnoid space, but also due to the host’s inflammatory response to this invasion [36]. The only widely researched agent that can limit subarachnoid inflammation is dexamethasone. The recommendation for the use of dexamethasone in bacterial meningitis unfortunately cannot be generalized and depends on the causative organism and the ability to administer dexamethasone within 1–12 h of administration of antibiotics [27,40,44,97].

In infant and children with Hib meningitis, administration of dexamethasone with antibiotics has shown a significant reduction in neurological sequelae rate (17%) compared to antibiotic only (23.4%) [27,59]. Additionally, rates of hearing loss were lower with dexamethasone use (12.9%) compared to antibiotics only (17.4%) [59,96,98,99]. In a large study done in Malawi, children with *H. influenzae* meningitis who received dexamethasone were less likely to have neurological sequelae compared to a placebo group (27% vs. 40%) [13]. Therefore, it is generally recommended that dexamethasone be administered before or with the first dose of antibiotics [1,13,96] when *H. influenzae* is confirmed or strongly suspected.

On the other hand, the use of dexamethasone in infants and children with pneumococcal meningitis is controversial as it has not been clearly proven to change the outcome [1,13,36,98,99]. Additionally, the use of dexamethasone for meningococcal meningitis was not proven to be effective in reduce neurological sequelae [96].

Better understanding of specific microbial and host factors contributing to CNS infection and inflammatory response may help in identification of new therapeutic targets and specific immunomodulatory regimens.

## 7. Long-Term Follow Up

All children who are diagnosed with meningitis should have a hearing assessment done before discharge or 1 month within discharge; even if hearing loss is not clinically suspected [13,61]. It is critical to perform audiology assessment month after diagnosis or earlier if possible, as up to 90% of children’s cochlea with hearing loss due to meningitis can ossify, preventing appropriate treatment with cochlear implants [100,101]. 

Children with seizure disorders require antiepileptic medication and should ideally have long-term follow up by a neurologist [7,8,67,70]. Children with hearing loss and/ or intellectual disability will need neurodevelopmental follow up and support for speech, language and social development. Mental health concerns and psychiatric problems are likely underreported in children who were diagnosed with meningitis and periodic mental health assessments by an appropriate specialist should be included into their long-term care [64].

## 8. Conclusions

Children with bacterial meningitis are at risk of developing neurological complications that include focal neurological deficits, subdural effusion, hearing loss, cognitive impairment, seizure disorder, and hydrocephalus. There is a need to optimize utilization of available vaccines and to develop vaccines for pathogens implicated in neonatal meningitis (GBS and *E. coli*). For children diagnosed with bacterial meningitis, starting antibiotic therapy without delay is critical for a good prognosis and to reduce the risk of developing neurological complications. So far, steroids are the only drug that can control inflammatory response, but effectiveness is limited to specific situations.

## Figures and Tables

**Figure 1 microorganisms-09-00535-f001:**
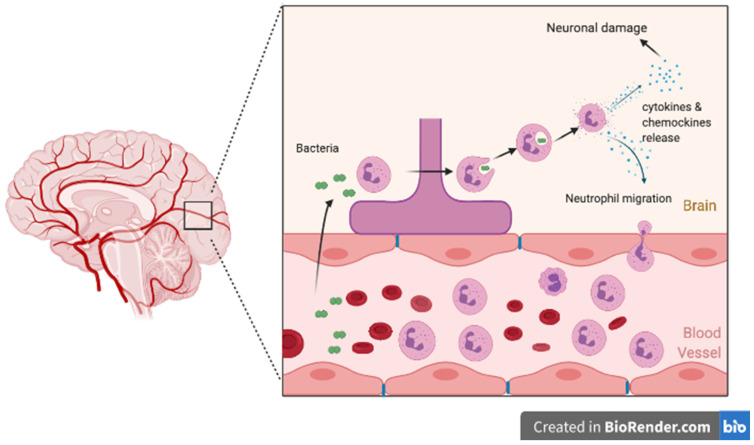
Pathophysiology of Neuronal Damage due to Bacterial Meningitis.

**Table 1 microorganisms-09-00535-t001:** Most common organism for different Age Groups.

Age Group	Most Common Organisms	References
Pre-term neonate	*Escherichia coli*, GBS *	[12,21,22]
Term neonate and infants < three months	GBS *, *E. coli*, *Streptococcus pneumoniae*, *Listeria monocytogenes*	[12]
Children ≥ three months to ten years	*S. pneumoniae*, *Neisseria meningitidis*, *Haemophilus influenzae* type b	[3,13,23,24]
Adolescent until 19 years old	*N. meningitidis*, *S. pneumoniae*	[4,25,26]

* GBS: Group B Streptococcus.

**Table 2 microorganisms-09-00535-t002:** Long and Short-term Neurological Complications following pneumococcal and meningococcal meningitis in Low and Middle Income Countries (LMICs) and High Income Countries (HICs).

	Pneumococcal Meningitis		Meningococcal Meningitis		References
	LMICs	HICs	LMICs	HICs	
Focal deficits	12%	3–14%	2–4%	3%	[13,39,40,41,42]
Hearing loss	25%	14–32%	19–23%	4%	[10,13,43,44,45]
Seizures	45–63%	15–48%	17–33%	2%	[40,45,46,47]
Cognitive impairment	4–41%	N/A *	4%	12–19%	[10,13,43,48,49]

*: Not avilable.

**Table 3 microorganisms-09-00535-t003:** Risk factors for developing neurological complication in Bacterial Meningitis.

Risk Factor	% with Neurological Complications	References
Young Age (infants < 12 months)	71%	[9]
Etiology: *S. pneumoniae*	75%	[9]
Altered Level of Consciousness on Presentation	82%	[9]
Delayed Presentation	N/A *	[6,9]
Delayed initiation of antibiotics	N/A *	[76]

* N/A: Not available.

## Data Availability

Not applicable.

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
