# Peer review of "Bacterial Meningitis in Children: Neurological Complications, Associated Risk Factors, and Prevention"

_microorganisms, 2021, doi:10.3390/microorganisms9030535_

Round 1
Reviewer 1 Report
This manuscript by Abdulwahed Zainel et al. describe interesting data on bacterial meningitis, especially focused on children.
This article is well-written and interesting but the subject remains very extensively describe.
Several typo remains to be corrected.
Note that numbers under 11 have to be written in full letters. "i.e" have to be italicized.
Reviewer 2 Report
The review - Bacterial Meningitis in Children: Neurological Complications, Associated Risk Factors, and Prevention - is well written and I enjoyed reading it. There are a few minor changes that need to be made:
There is no table 3 in the document, therefore table 4 needs to be reassigned to table 3.
Minor typos:
line 39- change is to in
Line 115- my should be may
line 151- fewer is misspelled
line 160 - anxiety is misspelled
